# Design and Impact of Grid Tariffs

Christian Winzer *[ID] and Patrick Hensler-Ludwig

ZHAW School of Management and Law, Center for Energy and Environment, 8400 Winterthur, Switzerland
* Correspondence: christian.winzer@zhaw.ch

**Abstract:** We propose a novel grid tariff design proportional to grid load and analyze its performance in comparison to other grid tariff designs with regards to (i) effectiveness, (ii) efficiency, (iii) profitability of technologies and (iv) equity. In the case of a large share of automated loads, time-of-use tariffs and critical peak prices create problematic new rebound peaks. Direct load control and capacity prices can reduce grid load without rebound peaks but are less effective at reducing both grid and energy costs. The novel tariff design proportional to the grid load can reduce both grid and energy costs but needs to be designed appropriately to avoid rebound peaks. Tariff impacts on the profitability of different technologies are more pronounced than equity impacts because households from all income brackets may be equipped with PV and flexible technologies.

**Keywords:** grid tariff; demand response; automatic load-control

## 1. Introduction

The switch to decentralized renewable energies and the increasing electrification of various economic sectors (e.g., mobility, heating) call for expansion in many power grids. The need for additional grid capacity depends heavily on installed technologies and user behavior which, in turn, is influenced by the incentives resulting from grid tariffs. However, the development of demand response (DR) as an alternative to grid expansion is currently limited due to the chicken-egg problem explained in [1]. As current levels of DR are low, grid operators tend to over-size the grid to prevent bottlenecks. This, in turn, means that there is a limited need to develop DR beyond today's levels. Efficient grid tariffs that minimize the sum of dispatch and investment costs for the total system, including the grid and flexibility options, could resolve this problem. By ensuring that the grid tariffs that prosumers pay correspond to the grid costs that result from their consumption behavior, efficient grid tariffs could incentivize the development of local flexibilities.

Most electricity consumers pay a regulated tariff for their electricity consumption. As illustrated in Figure 1, the electricity bill of households in Europe includes costs, ranging from the cost of electricity production ("Energy"), transmission and distribution grids ("Network"), cost of renewables ("RES") and different types of taxes ("Taxes" and "VAT").

Within this paper, we focus on the efficient design of tariffs for the recovery of grid costs. However, as we will discuss the same principles could also be applied to other cost components.

The three main tariff approaches that are used are lump sum payments (per time period), energy charges (per kWh) and capacity charges (per kW and time period). As illustrated in Figure 2, a variety of different combinations of these charges are used in EU countries. In addition to that, the share of costs that are allocated to each component, and the detailed design for each of these charging components vary greatly among different countries.

The great variety in approaches and designs raises the question regarding optimal tariff design. As a result of the increasing electrification of transport and heat sectors, short-term elasticity of demand is rapidly growing. At the same time, the rollout of smart meters is providing new opportunities for the smart control of these devices. Future-proof

tariffs should thus be designed in a way that leverages these opportunities to send optimal dispatch and investment signals.

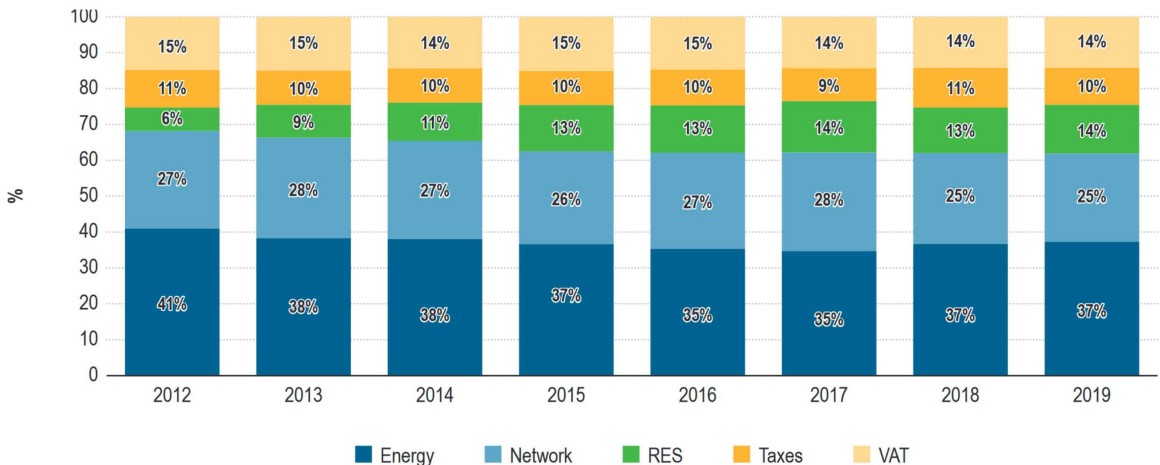

**Figure 1.** Composition of household electricity prices in Europe. Source: [2].

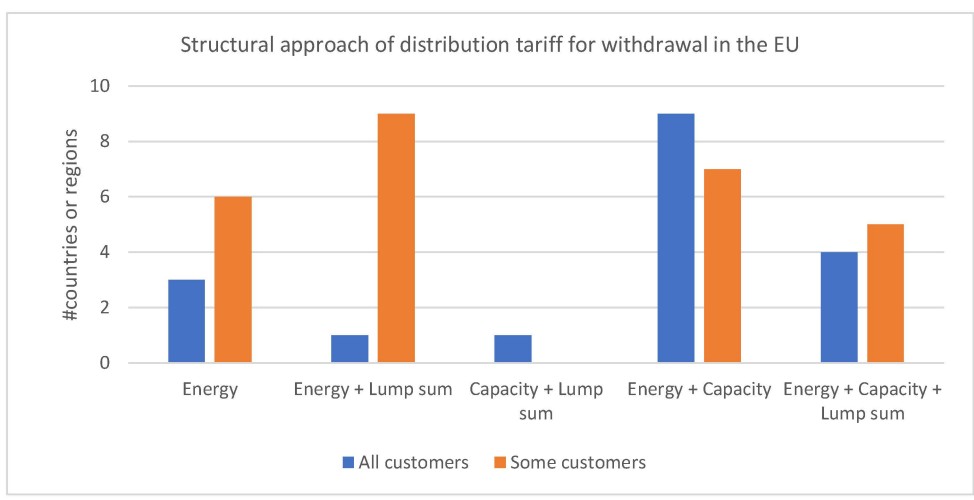

**Figure 2.** Number of countries using different tariff structures in the EU. Based on: [3].

Analyses of grid tariff designs are often limited to the tariff impact on grid peak load and the resulting grid investment need. Within this paper, we include an assessment of additional criteria such as the tariff impact on electricity generation cost, profitability of different technologies and cost-redistribution between households from different income brackets, which are also relevant for the choice between different tariff approaches.

The structure of this paper is as follows. In Section 2, we summarize the approaches that were suggested and the main relevant findings from the literature on optimal tariff design. In Section 3, we describe the tariff scenarios and in Section 4 the data used in the analyses for this paper. The results of our analysis are reported in Section 5 and discussed in Section 6. In Section 8 we describe our conclusions and policy implications.

## 2. Literature

To date, a substantial body of literature has explored different optimization strategies for home energy management systems (HEMS), particularly for appliance scheduling. Many decentralized approaches use linear optimization to determine households' responses to external price signals. Ref. [4] uses mixed-integer linear programming (MILP) to minimize household electricity costs by adjusting appliance load profiles in a time-dynamic pricing scheme. They model prosumer households with various adaptable appliances and

devices such as EVs, ESS, thermal storage systems, and household appliances. Ref. [5] minimizes electricity bills over a multi-day time horizon, incorporating a penalty term into the objective function to account for the prosumers' comfort. In addition to time-dynamic electricity tariffs, [6] employ MILP to investigate prosumers' short-term adaption under five distinct electricity tariffs, which include a capacity tariff and multiple bidirectional tariffs. They find that these two concepts significantly reduce grid interaction. Ref. [7] introduces a multi-objective optimization problem to study the balance between minimizing electricity costs and enhancing households' comfort under two different power-based electricity contracts.

Taking into account that individual optimal appliance scheduling is not necessarily beneficial to the grid operation, several studies propose frameworks that coordinate DR. Ref. [8] uses a bi-level optimization problem to investigate coordinated DR. At the lower level, they optimize the electricity bills of households, while in the upper level, they minimize the overall system load. Ref. [9] proposes an optimization approach at the neighborhood level to coordinate DR.

One of the main findings from this literature is that with increasing levels of automation, discontinuous price signals, such as time-of-use tariffs and critical peak prices could not reduce the peak load, and lead to an even higher rebound peak at the beginning of low-price periods. The solutions proposed to this problem include the introduction of a protection period after the high-tariff hours [10], the introduction of different price signals for different consumer groups [11] and the imposition of binding power limits during scarcity [4,9,12]. However, while the resulting dispatch may be effective in achieving a lower peak demand, these studies do not distinguish between the tariff approaches for recovering energy cost, grid cost or residual cost, and do not analyze how the suggested proposals affect other tariff design criteria, such as cost-recovery, welfare, equity and other dimensions of acceptability (i.e., fairness, transparency) [13–15]. In addition to that, only two of the studies assess the suitability of capacity tariffs [6,7], and none of them evaluate the impact of using a continuous tariff signal, instead of a discrete step-function.

In our paper, we will fill this gap, by exploring the impact of both a capacity tariff and a novel continuous tariff signal proportional to grid load and compare their performance to the performance of other approaches for avoiding rebound peaks along all the above-mentioned tariff design criteria and calculate the overall efficiency of tariffs including impacts on grid peak-load, grid and energy cost, technology profitability and equity.

## 3. Methodology

To simulate the behavior of consumers and prosumers in response to dynamic electricity prices and to analyze the effects of the resulting load profiles on the grid infrastructure and consequently grid costs, we use a linear optimization model. An overview of the model structure is provided in Figure 3. The inputs for this model include the "infrastructure scenario" and the "tariff scenario".

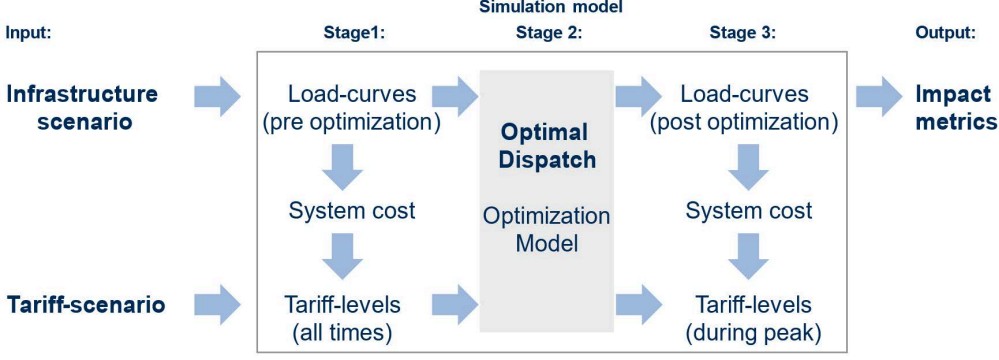

**Figure 3.** Structure of the simulation model.

The "infrastructure scenario" outlines the number, types, and equipment of households being simulated. We define 13 distinct consumer and prosumer types based on variations in asset equipment and household characteristics. Consumers can have different combinations of heat pumps (HP), electric vehicles (EV), or neither of these assets, while prosumers are additionally equipped with photovoltaic (PV) systems and potentially Battery Electric Storage Systems (BESS). We model BESS, EVs and HPs as shiftable in time since these devices offer the required flexibility and are likely to be the largest contributors to demand peaks by 2035 [16]. Other household loads, such as wet goods, stoves, consumer electronics, etc., are considered inflexible in the model.

We will analyze the effects of nine different "tariff scenarios" by comparing two distinct incentive mechanisms: capacity tariffs (type 2) and time-differentiated tariffs (tariff type 3–8) which are price-based mechanisms, along with a "direct load control" scheme (tariff type 9) which operated as an incentive-based mechanism.

1. flat: constant price per kWh during all hours
2. capacity: constant price per kW individual annual consumption peak
3. tou: time-of-use tariff, with a daily high- and low tariff period
4. cpp-h: critical-peak-price tariff, with a high tariff during those hours when load reaches its annual maximum and a low tariff otherwise
5. cpp-d: critical-peak-price tariff, with a high tariff during all high-tariff hours of the tou tariff on days, when load reaches its annual maximum and a low tariff otherwise
6. grid load: a tariff that is proportional to the projected grid load prior to load shifting
7. spot: a tariff that is proportional to spot prices
8. grid load and spot: a combination of 5 and 6.
9. dlc: direct load control tariff

The simulation model consists of three stages. In the first stage, system costs are calculated based on the exogenous infrastructure scenario. The electricity tariff levels are calibrated following the cost recovery principle which ensures that grid, energy and residual costs are fully recovered by each tariff scheme. The second stage models households' operational decisions on the electricity tariffs using a linear optimization model. After calculating the optimal household dispatch, the third and last stage of the model adjusts tariff levels according to the optimized load profiles to ensure grid cost recovery. Recalibrated tariff levels are then used as input for calculating the impact metrics. The mathematical notation and further details regarding the dispatch model, the method for tariff calibration and calculation of the impact metrics are provided in the Appendices A–E.

## 4. Data

The model was utilized to simulate a synthetic subset of a distribution grid, encompassing a single transformer station that serves 300 households. These households exhibit variations in terms of device equipment and consumption patterns. The simulation is conducted for two distinct years, 2020 and 2050, with an hourly resolution. Synthetic load profiles were employed for households, electric vehicles, heat pumps, and photovoltaic systems.

The simulation model is calibrated for a case study in the canton of Aargau, Switzerland. Within the model, two complete years are simulated with hourly resolution, differing in the distribution of flexible electrical appliances and installations. The electricity tariffs are aligned with the average electricity cost proportions from the year 2021, corresponding to an electricity cost of 20.01 Rp./kWh. Between the two simulated years, household appliance equipment varied based on survey data and forecasts. For the 2020 appliance equipment, data from the Swiss Household Energy Demand Survey (SHEDS) conducted from 2016 to 2020 were utilized. The survey encompassed 5000 Swiss households and gathered information about their appliance equipment and consumption behavior. Appliance equipment estimation for the year 2050 was performed using forecasts from the Swiss Federal Office of Energy [17]. These forecasts expect that by 2050 around 68% of all households will have a heat pump, 78% an EV, 47% a BESS and 67% a PV module. Household counts were estimated by adjusting household counts for each type from 2020 in a way that achieves

this marginal diffusion of the different technologies while minimizing the sum of squared deviations between the household share in 2020 and the household share in 2050.

The resulting number of household types for each scenario year and node is presented in Table 1. We differentiate between 12 different types of households, depending on whether the household is equipped with HP, EV, BESS or PV. For example, for 2020 we assume that 1 of the 300 households (0.3%) which we simulate is equipped with HP, EV and PV (type9), while during 2050 we assume that this number rises to 60 households (20%).

**Table 1.** Overview of household types and technologies.

| HH-Type | Household Features | | | | Household Count | | Household Share | |
|---|---|---|---|---|---|---|---|---|
| | HP | EV | BESS | PV | 2020 | 2050 | 2020 | 2050 |
| type1 | 0 | 0 | 0 | 0 | 211 | 66 | 70.3% | 22.0% |
| type2 | 1 | 0 | 0 | 0 | 57 | 0 | 19.0% | 0.0% |
| type3 | 0 | 1 | 0 | 0 | 5 | 29 | 1.7% | 9.7% |
| type4 | 0 | 0 | 0 | 1 | 4 | 0 | 1.3% | 0.0% |
| type5 | 1 | 1 | 0 | 0 | 3 | 4 | 1.0% | 1.3% |
| type6 | 1 | 0 | 0 | 1 | 3 | 0 | 1.0% | 0.0% |
| type7 | 0 | 1 | 0 | 1 | 0 | 1 | 0.0% | 0.3% |
| type8 | 0 | 0 | 1 | 1 | 8 | 0 | 2.7% | 0.0% |
| type9 | 1 | 1 | 0 | 1 | 1 | 60 | 0.3% | 20.0% |
| type10 | 1 | 0 | 1 | 1 | 6 | 0 | 2.0% | 0.0% |
| type11 | 0 | 1 | 1 | 1 | 1 | 0 | 0.3% | 0.0% |
| type12 | 1 | 1 | 1 | 1 | 1 | 140 | 0.3% | 46.7% |
| Total | | | | | 300 | 300 | 100% | 100% |

We assume that only the load from HPs, EVs or BESSs responds to flexibility incentives. The remaining portion of residential load and PV system generation is considered non-flexible.

Simulations were based on synthetic load profiles from the LoadProfileGenerator (LPG) [18]. PV production profiles and heat pump load profiles were calculated based on building-level data from a distribution grid operator in the canton of Aargau. Further details on the load profiles can be found in the Appendix F.

## 5. Results

This section presents the findings of the case study. First, in Section 5.1, we display the model results for the current scenario, where 28% of households possess one of the flexible technologies. Additionally, 8% of households in this scenario are equipped with a PV system. Moving on to the 2050 future scenario, the share of prosumers in the grid area rises to 67%, while the share of households with flexible technologies increases to 78%. The outcomes of this scenario are elaborated upon in Section 5.2.

### 5.1. Scenario 2020

#### 5.1.1. Effectivity

We assess the effectiveness of the tariffs based on the change in peak load incentivized by each tariff compared to the load under a flat tariff (Status Quo—SQ) within the considered network area. Figure 4 illustrates the change in load at the time when the peak load occurred in the SQ scenario (blue) as well as the change in the overall peak load (red).

The time-variable tariffs with two distinct price levels (TOU, CPP_h, and CPP_d) effectively decrease the load at the time of the peak load in SQ, but they result in a new peak load occurring at a different time, which is even higher than the original peak load.

The effect of increased peak loads due to new tariff structures is recognized in the existing literature as a "rebound peak" [11,12]. This effect stems from the step-like nature of price signals in time-varying tariffs with fixed, predefined price levels. This leads to a cumulative shift in flexible loads from high-tariff periods to the first subsequent low-tariff time slot. As these subsequent times often exhibit high load levels in comparison to the daily average, the shifted load can result in a peak load surpassing the original magnitude (Figure 5a).

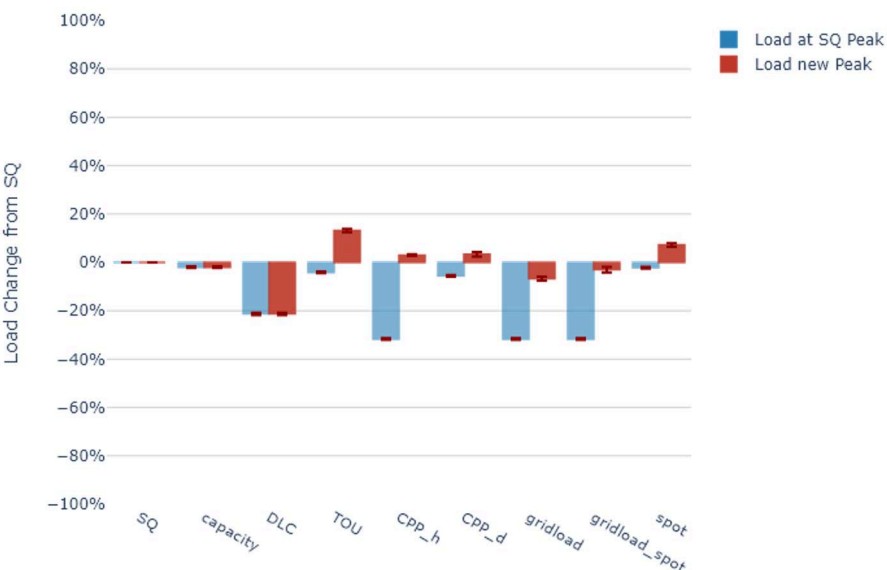

**Figure 4.** Change of system peak load compared to the SQ scenario in 2020.

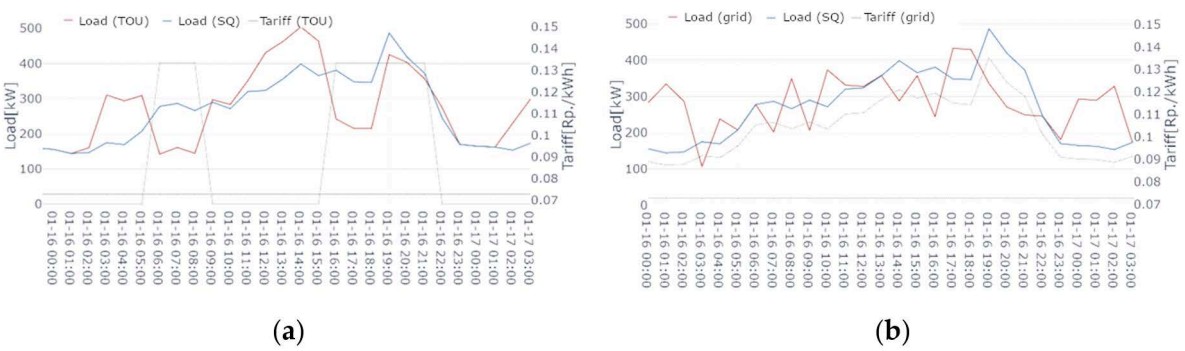

**Figure 5.** Transformer load in SQ scenario (blue line) compared to grid tariff (grey line) and corresponding transformer load (red line) under (**a**) TOU and (**b**) grid-load tariff.

Capacity tariffs (capacity) and direct load control (DLC) can avoid the rebound peak. However, the capacity tariff only achieves very small peak-load reductions of a few percentages, as the peaks of individual customers often do not coincide with the peak load of the total grid. The DLC tariff obtains the strongest peak-load reduction, as it assumes a central optimization with the objective of minimizing the peak load. In the 2020 scenario, the DLC tariff achieves a peak load reduction of around 20%. However, because of unbundling restrictions, DLC typically does not account for the energy costs as part of the optimization. Conceptually, it is, therefore, less suitable for minimizing the sum of grid and energy costs (see next section).

The tariff signal proportional to total load (grid load) achieves a lower peak-load reduction than DLC. However, in the 2020 scenario, it can avoid rebound peaks as it prompts flexible loads to shift to a period when the system load is comparatively lower (Figure 5b). Compared to capacity tariffs and DLC, an important advantage of the grid-

load tariff is, that it may be combined with an energy component that is proportional to spotprices (gridload_spot). We, therefore, expect that if the grid and energy tariff components are calibrated appropriately, this tariff should, therefore, minimize the sum of grid and energy costs (see next section below).

### 5.1.2. Efficiency

Figure 6 shows the sum of grid, energy and residual cost which result under each tariff for the 2020 scenario. Across the board, the price impacts of all tariffs remain relatively modest, ranging from a reduction of −3.2% to an increase of +1.6%. The two main reasons for this are the small number of households that are equipped with flexible technology, and the assumption, that only 30% of grid costs depend on peak load.

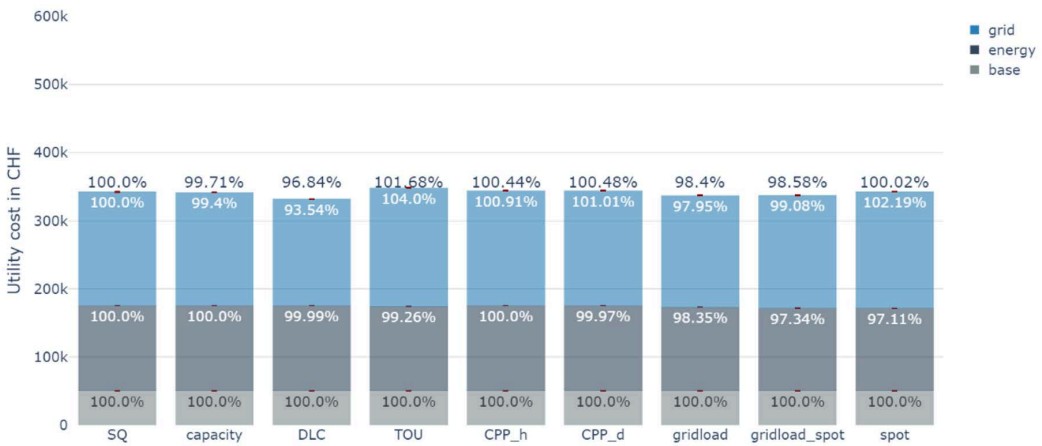

**Figure 6.** Grid, energy, and residual cost in the Scenario 2020.

As expected, the trend in network costs closely mirrors the outcomes of the effectiveness analysis, with DLC and grid-load tariffs reducing grid costs most strongly (by 6.46% and 2.05%, respectively).

Regarding the impact on energy cost, as expected, most tariffs hardly reduce the energy cost because of the high tariff periods for time variable tariffs (TOU, CPP_h, and CPP_d) cover hours with a high grid load, which may not be the hours with the highest spot prices and capacity tariffs (capacity) and direct load-control does not target a reduction in energy cost. Tariffs proportional to the spot price (spot), therefore, achieve the strongest reduction in energy cost (by 2.89%), followed by the combined tariff where the grid component is proportional to the total load, and the energy component is proportional to spot prices (gridload_spot).

However, the result regarding the sum of all cost components is different from expected. Even though the combined gridload_spot tariff is the only one that explicitly considers both cost components, the total cost reduction achieved by this tariff (1.42%) is lower than the total cost reduction achieved by direct load control (3.16%) or the grid-load tariff (1.6%). We assume that this is due to a sub-optimal design of the grid-load tariff signal, which puts too much weight on a reduction in grid-load during hours when the grid is not congested. This will be further discussed in Section 6.

### 5.1.3. Profitability

Figure 7 displays the average electricity costs for households with different appliances under the baseline scenario with a constant price per kWh.

Figure 8 illustrates how electricity costs for households equipped with different devices and appliances change under different tariff scenarios compared to the baseline scenario with a constant price per kWh. With capacity pricing, electricity costs rise for households lacking flexible technology. The most significant increase occurs in households with PV systems. This is due to different approaches to allocating variable grid costs. In the baseline

tariff, variable grid costs are evenly distributed across total energy consumption. Since households with PV systems exhibit lower grid consumption due to self-consumption, they bear only a minor share of variable grid costs under a fixed tariff. However, under the capacity tariff, variable grid costs are allocated solely to individual peak loads, resulting in higher electricity costs for PV-equipped households. The introduction of additional flexibilities can mitigate this cost increase to some extent, but the flexibility of EVs and BESS is not enough to completely offset the rise in costs, resulting in higher electricity expenses for households with these device combinations as well.

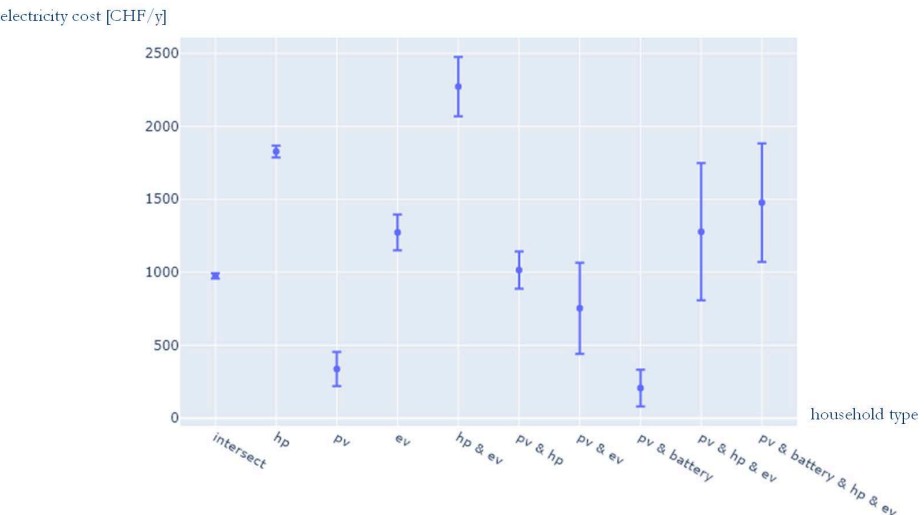

**Figure 7.** Electricity cost in case of a constant price per kWh in the Scenario 2020.

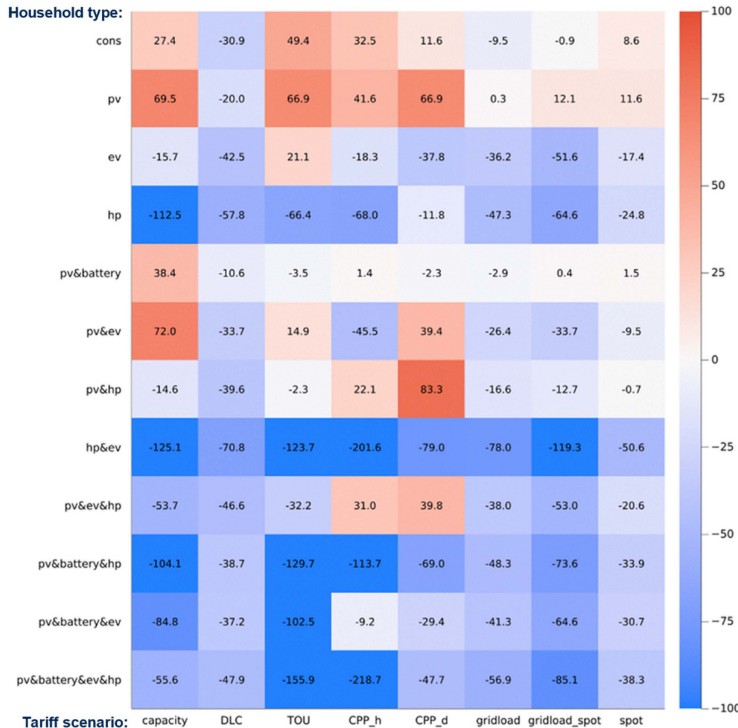

**Figure 8.** Difference in electricity costs [CHF/year] compared to baseline tariff in 2020.

This phenomenon is also observed under TOU, CPP-h, and CPP-d. These tariffs spread variable network costs over a limited number of hours, mainly in the morning or evening when PV generation is lower. Conversely, households with high electricity consumption, which bear a significant portion of variable grid costs in the baseline scenario, can benefit

from these tariffs by reducing their peak load or shifting consumption away from high-tariff periods. HPs are particularly advantageous as they can offer flexibility throughout the cold season. EVs, due to their limited stationary time, exhibit less flexibility, and therefore, have a smaller impact on load shifting.

Similarly, the spot tariff leads to increased annual electricity costs for prosumers and non-flexible households. For non-flexible households, this is primarily due to the low efficiency of the spot tariff, leading to an overall cost increase. In the case of prosumers, the strong negative correlation between spot prices and the PV generation profile results in lower savings from self-consumption during midday hours in the summer. Since this effect is more pronounced in the summer months, additional flexibility in the form of HPs has a modest cost-reducing effect. EVs, on the other hand, provide a stronger cost-reducing impact for prosumers under this tariff. In the gridload_spot tariff, the negative correlation between PV generation and price signal persists, causing prosumers without additional flexibilities to fare worse under this tariff compared to the baseline. Inflexible consumers, however, benefit from the tariff's efficiency, resulting in only minor additional costs, typically less than 1 CHF per year.

Under the grid-load tariff, the additional costs for prosumers and households without flexibility are notably diminished. Only households with PV systems experience a slight uptick in electricity costs. This phenomenon can be attributed to both the effectiveness and efficiency of the grid-load tariff in the 2020 scenario, which lowers the costs for allocation. Furthermore, variable grid costs are still assigned to the total annual energy consumption, albeit with grid load-dependent weighting. Consequently, the cost increase for prosumers is less pronounced compared to time-variable or capacity tariffs.

Remarkably, the DLC tariff is the sole tariff resulting in decreased annual costs across all household types. This is due to the substantial reduction in total costs, which persist despite the fixed allocation to energy purchases.

### 5.1.4. Equity

Results in the previous section demonstrated that households equipped with multiple flexible appliances tend to profit more from dynamic tariffs than households with less flexibility. In order to verify, to what extent this effect will lead to a cost redistribution from richer to poorer households, Figure 9 shows the average electricity price for households from different income brackets under each of the electricity tariffs.

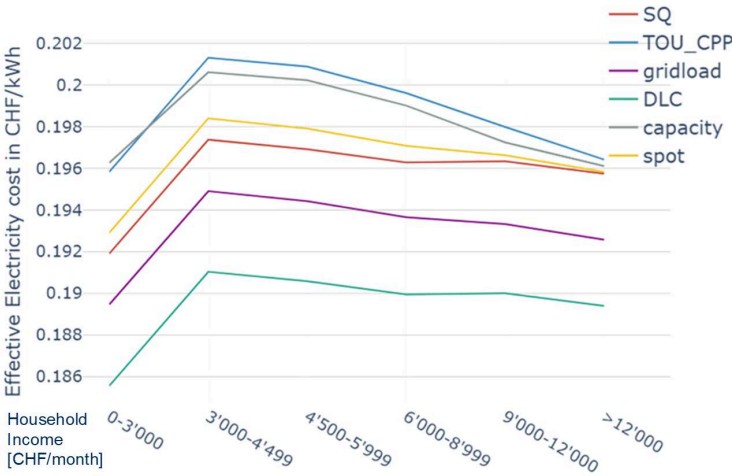

**Figure 9.** Electricity costs per Household income group 2020.

Contrary to our expectations, the main impact of the tariffs is a parallel shift of the average cost across most income brackets. The reason for this is, that in 2020 heatpumps are the only flexible technology which is available to a larger number of households, and the share of households from different income brackets in Table 1 is roughly the same in

the case of "type 2" (with heatpump) and "type 1" (without flexibility). Surprisingly, the households with the lowest income in this case study exhibit the lowest electricity costs under all tariff scenarios. This is due to the fact, that there are quite a few households from the lowest income group which are equipped with PV panels, e.g., because they are tenants of a landlord that chose to install PV.

*5.2. Scenario 2050*

5.2.1. Effectivity

In the 2050 scenario, the peak load rises from 492 MW in the 2020 scenario to 919 MW in the 2050 scenario because of electrification. However, many of the new loads are flexible, so the share of households with flexible technology increases to 78% and 67% of the households are assumed to have PV. The impact of the different tariffs on peak load is illustrated in Figure 10.

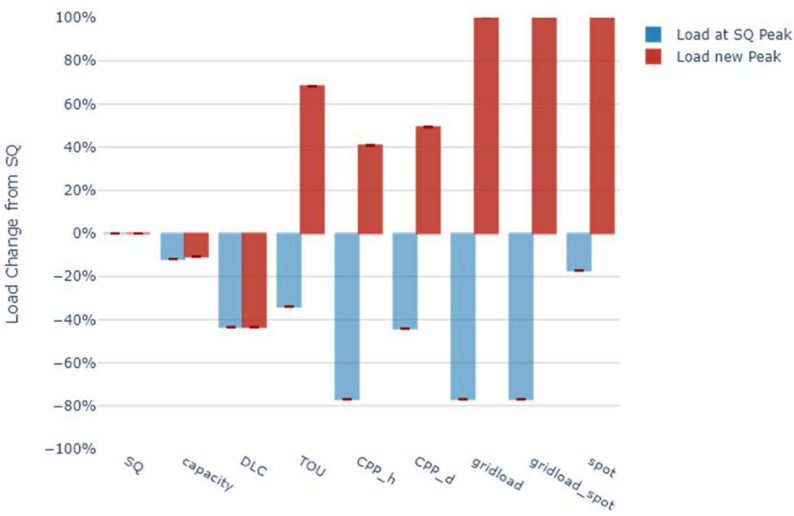

**Figure 10.** Change of system peak load compared to the SQ scenario in 2050.

As a result of the higher share of flexible loads, the rebound peak which is created by time-varying tariffs (TOU, CPP_h, and CPP_d) increases sharply from between 5% and 15% in 2020 to more than 40% in 2050.

Capacity tariffs and DLC can effectively avoid rebound peaks, and reduce grid peak load by about 10% (in the case of capacity tariffs) and more than 40% (in the case of DLC). However, even this load reduction is inadequate to bring the total peak load down to the levels observed in the 2020 scenario, necessitating grid expansion to accommodate the increased demand.

By contrast to the 2020 scenario where the grid-load tariff contributed to a reduction in the peak load, it can no longer avoid rebound peaks in the 2050 scenario. Surprisingly, the rebound peak, which is caused by grid-load tariffs, even exceeds that which is caused by time-varying tariffs, causing the peak load to more than double compared to the baseline tariff. This phenomenon is a result of the shape of the price function associated with the grid-load tariff. Specifically, there is one single time step per day featuring a minimum price. Consequently, the entire flexible load around this time step is shifted to the moment with the lowest price. Given that these low-price periods usually occur during nighttime hours when a substantial number of EVs are available for charging (as depicted in Figure 11), the charging load of multiple EVs is concentrated into a single time frame. This concentration of load results in significant rebound peaks. Suggestions for avoiding these rebound peaks will be discussed in Section 6. In contrast to grid-load tariffs, the timing of the high tariff periods for the other time-varying tariffs (TOU, CPP_h, and CPP_d) occurs during morning or late afternoon hours. During these times, the proportion of available EVs is relatively limited (Figure 11), which explains the lower rebound peaks.

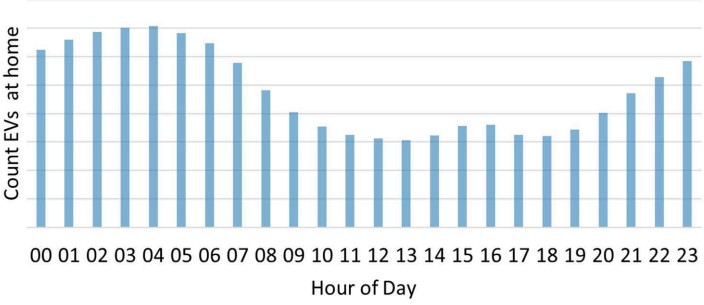

**Figure 11.** Histogramm of EVs at home.

### 5.2.2. Efficiency

Regarding grid costs, the same as for the 2020 scenario, the trend in network costs in Figure 12 closely mirrors the outcomes of the effectiveness analysis. While time-varying tariffs (TOU, CPP_h, and CPP_d, and in 2050 also the grid-load tariff) increase grid costs due to rebound peaks, capacity tariffs and DLC can reduce the grid cost by 3.6% and 11.32%, respectively.

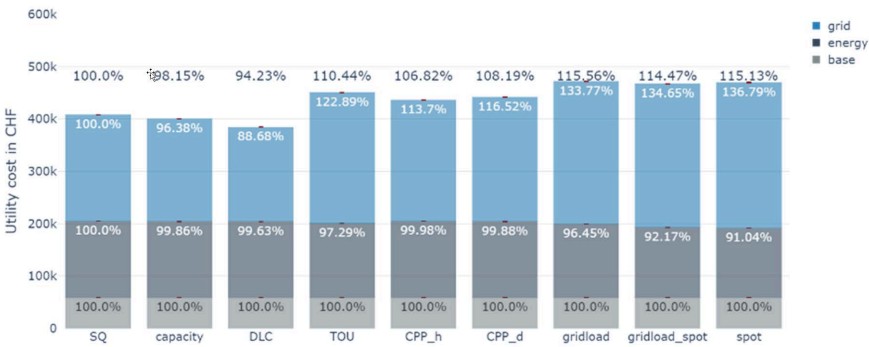

**Figure 12.** Grid, energy, and residual cost in the scenario 2050.

With regards to energy cost, as expected spot tariffs achieve the largest reduction in energy cost of about 9%, followed by the gridload_spot tariff, which reduces energy cost by about 8%.

Looking at the sum of all cost components, the impact of rebound peaks on grid costs clearly outweighs the beneficial impact of the grid load and spot tariffs on energy costs. Among the tariff designs that were investigated in this study, direct load control, therefore, achieves the largest cost reduction of about 12%, even though it does not significantly reduce energy costs. However, as we will describe in Section 6, we assume that further improvements to the gridload_spot tariff design could lower the total cost even further.

### 5.2.3. Profitability

The substantial rise in total costs associated with time-variable tariffs has a significant impact on a considerable proportion of households. As shown in Figure 13, time-varying tariffs increase annual electricity cost for most households. Only prosumers with PV & battery storage are partially shielded from escalating bills through potential reductions in their electricity consumption during peak periods, particularly with TOU and CPP tariffs. This is achieved by using their battery storage to offset demand during high-tariff periods, effectively mitigating the pronounced upswing in variable grid costs.

A noteworthy observation is that households equipped solely with an EV experience cost increases under the capacity tariff in the 2050 scenario, a contrast to the benefit they gained in the 2020 scenario due to decreasing prices in that tariff. While these households can curtail their peak load in the 2050 scenario, reducing it by an average of 29% from 13.4 kW to 9.5 kW, most households in the 2050 scenario possess multiple flexible appliances.

These households exhibit even greater peak load reductions, averaging at 58%, from 13.3 kW to 5.5 kW. As a result, they bear only a minor portion of the variable grid costs under the capacity tariff, leaving the less flexible households to shoulder a larger portion of these costs. In the 2050 scenario, this includes households with an EV or a PV system along with an EV, leading to a less favorable outcome for them compared to the baseline tariff. In contrast, the DLC tariff significantly diminishes variable grid costs, contributing to a reduction in electricity costs for all types of households.

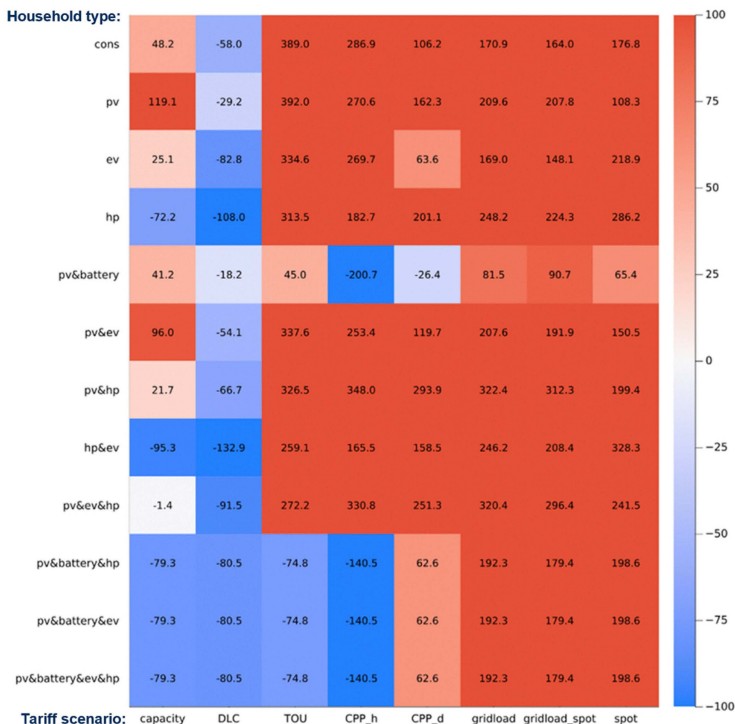

**Figure 13.** Difference in electricity costs [CHF/year] compared to baseline tariff in 2050.

### 5.2.4. Equity

Although we assumed that the income distribution within household types remains unchanged between 2020 and 2050, the impact of the tariffs on average electricity cost in 2050 (Figure 14) changes in two important ways compared to 2020.

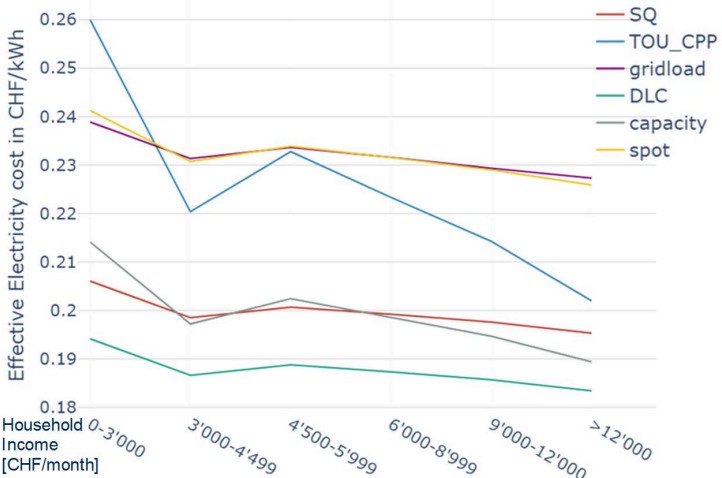

**Figure 14.** Electricity costs per Household income group 2050.

First, the overall magnitude of the changes in electricity prices increases sharply. This is caused by the fact that the larger share of households with flexible devices in 2020 versus 2050 (see Table 1) results in a much larger tariff impact on grid peak load (Figure 10) and as a result, a larger impact on grid cost (Figure 12).

Second, while the main impact of most tariffs is still a parallel shift, costs for poorer households increase significantly more than the cost for richer households in the case of capacity, TOU and CPP tariffs.

## 6. Discussion

The results from our simulations confirm the occurrence of rebound peaks in the case of time-varying tariffs in combination with automatic load control [11,12]. Direct load control and capacity-based tariffs can effectively avoid rebound peaks. However, due to the legal unbundling between grid and energy companies, direct load control is typically either used by grid operators to minimize grid costs or by energy suppliers, to minimize energy procurement costs.

Novel grid charges which are proportional to the grid load could overcome this problem. If customers are charged a gridload tariff (for their grid usage) and an energy tariff proportional to the spot price (for their energy consumption), they could ask independent aggregators to dispatch their loads in a way that minimizes the sum of grid and energy costs, which should minimize the sum of the total system cost.

While the performance of conventional direct load control (DLC) and capacity tariffs is similar in 2020 and 2050, the performance of the novel grid-load tariffs depends on the simulation year. Grid-load tariffs achieve the second biggest reduction in system peak load (Figure 4) and system cost (Figure 6) during 2020 when there are few flexible loads. This changes dramatically during 2050, when grid-load tariffs attract a much larger volume of flexible loads, resulting in rebound peaks of more than 100% (Figure 10) and increasing system cost (Figure 12). We believe that the failure of grid-load tariffs to minimize system cost, especially during years with a high share of flexible loads is due to the following two main reasons:

First, the tariff levels of the grid-load tariff were fixed in advance so that all flexible loads maximized consumption during the few time periods with the lowest tariff level. This could be avoided if tariff levels were fixed ex-post, based on the actual grid load which results after load-shifting and grid-load measurements were communicated close to real-time [19]. This would mean that the incentive to shift towards a certain time period would be reduced as more loads start to shift towards that time period. Creating such a feedback loop between actual grid load and tariff levels could effectively mitigate rebound peaks.

Second, the grid load tariff which we have currently tested created incentives to reduce load even during such periods, when the grid was not congested. While this may be desirable from a grid perspective, it may reduce the incentive to minimize energy costs. To avoid this, grid tariffs should only send incentives for load-shifting, when the grid is constrained. During all other time periods, the grid tariff should not incentivize load-shifting, so that flexibility has an incentive to dispatch in a way that minimizes energy cost.

We have summarized our ideas for designing a grid tariff that complies with these criteria in the following research note [20]. However, due to the complexity of modeling real-time tariffs, we have so far not been able to assess the impact of these proposals.

## 7. Limitations

When interpreting the results, it is important to note that certain simplifications and assumptions were made in this model.

First, we assume automatic load control of all flexible loads in response to electricity prices. This assumption is clearly not met today. However, as the costs for automation fall, we expect that more and more households will be equipped with automatic load control.

Second, the determination of grid costs is based on historical data and greatly simplifies the grid-cost impact that will occur in a real grid. In particular, we assume a linear

dependency between grid peak load and grid costs, whereas in practice, the impact will follow a step-function.

Third, our model simulates only a single electricity tariff within each grid area. In a liberalized electricity market, households have the freedom to choose their tariff. Therefore, the extreme scenario of a single tariff will be unlikely to occur in a distribution grid.

Fourth, it was so far not possible to model the tariff designs suggested in [20]. Our current model either calculates the optimal dispatch of each household independently of each other or as part of a central optimization problem (see Appendix C). By contrast, the calculation of an ex-post grid tariff with near real-time load forecasts would require an iterative calculation of decentral optimizations by each household, which are linked to each other through their impact on the total grid load.

Fifth, in our current paper, we have simulated the impact of tariff designs for a single grid node. This could be generalized in subsequent papers by modeling tariff impacts on a system with several nodes and calculating nodal versions of the proposed tariff designs based on estimations of the system state [21].

Last but not least we assume exogenous wholesale prices in our model, implying no feedback effect of behavioral changes in the grid area on wholesale prices. While this assumption might hold true when new tariff schemes are introduced in a limited number of distribution grids, it may not remain valid in the case of a large-scale rollout.

## 8. Conclusions

The increasing penetration of electric vehicles, heat pumps, and energy storage, along with the rollout of smart metering technology, is expected to significantly increase the potential for residential demand response. Leveraging this potential is a critical challenge in future electricity markets. Price-based demand response strategies have the potential to play a pivotal role by providing incentives for households to adjust their electricity consumption patterns in a cost-effective manner. However, since loads are controlled by households, or aggregators who provide the load control as a service rather than the grid operator, price incentives need to be carefully crafted to ensure that the decentralized load adjustments reduce the need for grid expansion.

This research aims to contribute to the discussion on electricity tariff design by investigating the impacts of various tariff structures on the decisions of households equipped with different combinations of photovoltaics, batteries, electric vehicles, and heat pumps. The study employs four evaluation metrics: (i) effectiveness, (ii) efficiency, (iii) profitability of technologies and (iv) equity.

The first finding of our research is the importance of including appropriate impact metrics. For example, we find that equity impacts on households tend to be smaller than the impacts on the profitability of different technologies, because a non-negligible share of poor households may live in houses with PV production or flexible loads. Impacts on households with and without flexible technologies may thus not be a good proxy for the distributional implications of different policies. Likewise, an exclusive evaluation of the impact of grid tariffs on grid load may be misleading, as grid tariffs that achieve a lower reduction in grid load may still be preferable as they perform better at reducing total system cost.

Regarding grid tariff design, our findings confirm that time-varying network tariffs such as time-of-use (TOU) tariffs or critical peak prices (CPP) can lead to rebound peaks that surpass the original load peak by a significant margin.

Dynamic load control (DLC) by grid operators can avoid rebound peaks and reduce grid load by up to 43%. Capacity tariffs can also avoid rebound peaks but reduce the grid load to a much smaller extent (up to 3.6%), as individual consumption peaks do not correlate well with total grid consumption.

Due to unbundling regulations, grid operators in Switzerland and the European Union may not use direct load control to minimize energy procurement costs. Direct load control

by grid operators is, therefore, not a suitable option for reducing the sum of grid and energy costs in these countries, while it might be a viable option in other jurisdictions.

A novel grid tariff that depends on grid load could overcome both problems. If it is appropriately designed, it could incentivize customers to reduce grid load and avoid rebound peaks. At the same time, customers who are exposed to a grid load tariff (for their grid usage) and an energy tariff (for their energy consumption) should have an incentive to minimize total system cost.

To avoid rebound peaks, the grid-load tariff should depend on real-time grid load instead of ex-ante projections of grid-load. At the same time, the grid-load tariff should avoid load-shifting incentives when the grid is not constrained, so that flexibility can be used to minimize energy procurement costs.

**Author Contributions:** Conceptualization, methodology, validation, formal analysis, investigation, resources, project administration: P.H.-L. and C.W.; software, data curation, visualization, writing—original draft preparation: P.H.-L.; writing—review and editing, supervision, funding acquisition: C.W. All authors have read and agreed to the published version of the manuscript.

**Funding:** Most of this research was funded by the Swiss Federal Office of Energy (SFOE), Project Number: SI/501899-01. During article finalization and submission, Christian Winzer received funding by the Swiss Federal Office of Energy's SWEET programme as part of the project PATHFNDR. Open access funding provided by ZHAW Zurich University of Applied Sciences.

**Data Availability Statement:** Dataset available on request from the authors.

**Acknowledgments:** We want to thank our colleague Ingmar Schlecht, as well as our SFOE contact Wolfgang Elsenbast and other members of the advisory group for their helpful comments and the fruitful discussions during the NETFLEX and PATHFNDR projects.

**Conflicts of Interest:** The authors declare no conflicts of interest. The work was financed by the SFOE but the authors are solely responsible for the content and conclusions.

## Appendix A. System Cost Calculation

As the subject of investigation in our model is a synthetic grid infrastructure, estimating the total system costs becomes necessary. We quantify the initial system costs by valuing the load characteristics of the infrastructure scenarios using the price components of Swiss electricity prices in 2021 (the data were obtained from the Swiss regulator, which publishes an overview of Swiss electricity prices once a year on https://www.elcom.admin.ch/elcom/de/home/themen/strompreise/tarif-rohdaten-verteilnetzbetreiber.html). The sum of household loads l_(n,a) over one year is multiplied by the electricity price components:

$$C_{\text{system}} = \sum_{n=1}^{N} l_{n,a} * \left( p_e + p_g + p_t \right) \tag{A1}$$

where the electricity price is composed of an energy component p_e, a grid component p_g, and a residual component containing taxes and subsidies p_t. The values of these price components are displayed in Table A1.

**Table A1.** Median of Swiss household electricity prices for 2021 (Source: ElCom).

|  | Grid | Energy | Taxes, etc. | Total |
|---|---|---|---|---|
| Household type H4 [Rp./kWh] | 9.47 | 7.73 | 3.17 | 20.37 |

We assume that residual costs are independent of consumers' and prosumers' load profiles and should, therefore, be allocated via lump sum or a fixed electricity price component per connection point, following economic literature [22,23].

Regarding grid costs, we assume that 30% of the total grid costs depend on the grid peak load, and therefore, on consumers' load profiles, while 70% of the total grid

costs are driven by structural grid characteristics such as grid area, terrain, and topology, making them independent from consumers' load profiles [24]. The variable grid cost can be expressed as:

$$C_{g,var} = 0.3 * \max\left(\sum_{n}^{N} l_{n,t}\right) * C_g \qquad (A2)$$

This is indeed a simplification. Due to the lumpiness of investment decisions, the actual cost function is more likely to resemble a step function, where the total cost remains constant until peak demand surpasses capacity limits and necessitates capacity expansion. However, as elucidated in [25], the cost estimates are heavily influenced by the method used to model long-term marginal costs. Since the precise positioning and magnitude of cost steps are subjective, we assume the aforementioned linear function as an approximation for the genuine network costs.

Regarding energy costs, we assume that the marginal cost of energy production follows a pattern similar to the European spot price in 2018. We calculate the total cost of the energy that is needed to serve load $l_{n,t}$ for household n at time t as the product of the European spot price and the demand.

$$C_e = \sum_{t}^{T} l_{n,t} * p_{spot,t} \qquad (A3)$$

By employing this formula, we assume that infra-marginal rents, which producers receive whenever the price exceeds their marginal cost, serve as a reasonable proxy for their fixed production costs. As demonstrated by [26], in the context of an optimal production portfolio this assumption should hold true in the equilibrium situation.

## Appendix B. Tariff Calibration

We calibrate a separate tariff for each cost component (energy, grid, residual cost) to ensure the recovery of the respective total costs. We allocate the costs that are not directly influenced by the load profile of consumers as a fixed price per consumption unit. This includes residual costs such as renewable energy subsidies and taxes, as well as the fixed cost of the grid. Alongside this fixed price, we incorporate the costs directly influenced by consumers' load profiles (variable costs) in the form of marginal costs, according to the tariff approach. The variable costs contain the variable grid costs as well as energy costs. In the subsequent subsections, we outline the calibration function for each of the approaches that we integrated into our model.

### Appendix B.1. Flat Tariff

In the case of a flat tariff, the total cost that needs to be recovered is distributed equally over the load *l* during all hours $t = 1 \dots T$ of the year. The resulting tariff levels $tariff_{flat,t}$ at each time-step *t* can, therefore, be calculated as:

$$tariff_{flat,t} = \frac{C_{system}}{\sum_{t}^{T} l_t} \qquad (A4)$$

### Appendix B.2. Time-of-Use Tariff (tou)

In the case of a time-of-use tariff, the variable cost that needs to be recovered is distributed equally over the load l during all peak price periods $t \in T_{HT_{tou}}$ of the year. The fixed costs are allocated in the form of a fixed price across the total load throughout the year. The resulting tariff levels $tariff_{tou,t}$ at each time-step t can be calculated as:

$$tariff_{tou,\,t} = \begin{cases} \frac{C_{g,fix}}{\sum_{t \in T} l_t} + \frac{C_{g,var}}{\sum_{t \in T_{HT_{tou}}} l_t}, & t \in T_{HT_{tou}} \\ \frac{C_t + C_e + C_{g,fix}}{\sum_{t \in T} l_t}, & t \notin T_{HT_{tou}} \end{cases} \qquad (A5)$$

*Appendix B.3. Critical-Peak-Price during Selected Hours (cpp_h) and Selected Days (cpp_d)*

For critical peak pricing we differentiate between an hourly CPP-tariff and a CPP during selected days. In the hourly CPP-tariff, the peak price is only enforced during hours with the highest demand. The total cost that needs to be recovered is distributed equally across the load l during all high-tariff hours $t \in T_{HT_{cpp}}$ of the year. The daily CPP is applied during fixed hours of the day on the days with the highest demand. As to the TOU tariff, the fixed costs are allocated in the form of a fixed price across the total load throughout a year. The resulting tariff levels $tariff_{tou,t}$ at each time-step t can be calculated as:

$$\text{tariff}_{cpp,\,t} = \begin{cases} \frac{C_{g,fix}}{\sum_{t \in T} l_t} + \frac{C_{g,var}}{\sum_{t \in T_{HT_{cpp}}} l_t}, & t \in T_{HT_{cpp}} \\ \frac{C_t + C_e + C_{g,fix}}{\sum_{t \in T} l_t}, & t \notin T_{HT_{cpp}} \end{cases} \tag{A6}$$

**Tariff that is proportional to gridload (gridload)**

The grid-load tariff allocates the variable grid costs proportionally to the grid load. As in the other tariffs the fixed costs are allocated in the form of a fixed component per consumption unit. The tariff level $tariff_{gridload,t}$ is calculated as the sum of the fixed component and the grid load $l_t$ at time t multiplied by a constant proportionality factor:

$$\text{tariff}_{gridload,\,t} = \frac{C_{g,fix}}{\sum_t^T l_t} + l_t \cdot \frac{C_{g,var}}{\sum_t^T l_t^2} \tag{A7}$$

*Appendix B.4. Spot Price (Spot)*

While all previous tariffs allocated a flexible grid component to the consumer, the spot tariff assigns a flexible energy cost component to the consumer that is proportional to the spot price. The tariff level $tariff_{spot,t}$ is calculated as the spot price $p_{spot,t}$ at time t multiplied by a constant proportionality factor:

$$\text{tariff}_{spot,\,,\,t} = p_{spot,t} \cdot \frac{C_e}{\sum_t^T p_{spot,t} * l_t} \tag{A8}$$

*Appendix B.5. Capacity Price*

Additionally, alongside the volumetric tariff approaches, we examine capacity tariffs as another scheme that researchers have analyzed [6,7] for the purpose of recovering grid costs in accordance with the cost reflectivity principle. The capacity tariff allocates the grid costs over the sum of the maximum load of each household n as follows:

$$\text{tariff}_{capacity} = \frac{C_g}{\sum_n^N \max(l_{t,n})}, \quad t \in T \tag{A9}$$

*Appendix B.6. Direct Load Control (loadcontrol)*

In the case of direct load control, we assume that the consumers receive a flat tariff, as there is no need to incentivize load-shifting through price changes.

**Appendix C. Optimization Problem**

We model the demand response of consumers and prosumers to price-based and incentive-based tariff mechanisms using an optimization approach. For price-based tariffs, we introduce dynamic grid and/or energy components with the goal of reducing grid congestion. By altering tariff settings, we create incentives for load shifting, prompting households to respond by minimizing their electricity costs. This is simulated through a decentralized optimization model, assuming households respond rationally to exogenous tariff signals. In contrast, for incentive-based tariff mechanisms, a central optimization problem has been formulated to minimize congestion at transformer stations. Both approaches will be described in separate sections below.

*Appendix C.1. Decentral Optimization—Household Optimization Problem*

The decentralized optimization model calculates the load profiles for each household which result from an optimal dispatch under a given exogenous tariff scenario. The model is formulated as a linear optimization program (LP) that determines the cost-minimal operation of a prosumer's storage devices (A10). To address potential dissatisfaction caused by device scheduling, a penalty term is introduced into the objective function. In the case of heat pumps, this penalty term corresponds to a deviation from the desired ideal indoor temperature (A11). For electric vehicles, it corresponds to a deviation from the maximal state of charge that would have been achievable without flexibility provision at the departure time (A12). For a single household, this is expressed by:

$$
\min C_n = \sum_t^T (l_{t,n} * \Delta T * p_t^{buy} - i_t^{PV} * \Delta T * p_t^{sell} + C_t^{EV, HP_{Pen}}) + \max(l_{t,n}) * p^{power} \quad (A10)
$$

$$
C_{t,n}^{HP_{Penalty}} = \Delta SOC_{t,n}^{HP} * P_{LS}^{HP} \; \forall t \quad (A11)
$$

$$
C_{t,n}^{EV_{Penatly}} = \left( \left( SOC_{max,t}^{EV} - SOC_{t,n}^{EV} \right) * u_{t,n}^{EV,departure} \right) * P_{LS}^{EV} \; \forall t \quad (A12)
$$

with:

$l_{t,n}$ load from household n at time t

$i_{t,n}$ injection from household n at time t

$\Delta T$ Simulation time interval (1 h)

$p_t^{buy, \, sell}$ electricity price at time t/injection few at time t

$p^{power}$ capacity price-at time t

$C^{LS}$ Dissatisfaction term for load shifting

This objective function is subject to power balance constraints. The power consumption for a household at time t is described by (A13). Equation (A14) ensures that the amount of generated power from the PV plant is either consumed by the household or injected into the grid.

$$
l_{t,n} = l_{t,n}^{fix} + l_{t,n}^{EV,ch} + l_{t,n}^{HP,SQ} + l_{t,n}^{HP} - l_{t,n}^{PV,used} + l_{t,n}^{BESS} \; \forall t, n \quad (A13)
$$

$$
g_{t,n}^{PV} = i_{t,n} + l_{t,n}^{PV,used} \; \forall t, n \in N_{PV} \quad (A14)
$$

with:

$l_{t,n}^{fix}$ Not shiftable part of the load for household n at time t

$l_{t,n}^{EV, \, ch}$ Power demand to charge EV/BESS for household n at time t

$l_{t,n}^{HP}$ Power demand of the heat pump for household n at time t

$g_{t,n}^{PV}$ generation from PV for household n at time t

$l_{t,n}^{PV/BESS, \, used}$ Power consumed from PV/BESS for household n at time t

$i_{t,n}$ Power injected to grid from PV for household n at time t

The objective function is subject to various device-specific constraints that define the operation of flexible assets, namely BESS, EV, and HP. The operating limits of a BESS are defined by Equations (A15)–(A17), ensuring that the BESS operates within its state of charge (SOC) and power limits. Additionally, these equations ensure that the BESS is solely charged by the PV plant. The charging and discharging logic of a BESS is stipulated by Equation (A18). Electric vehicles are equipped with battery electric storage systems that are discharged while driving and available for charging, thereby providing flexibility during parking times. Consequently, the constraints for BESS and EV share many characteristics. In addition to the constraints describing the BESS operation, the EV model incorporates an extra variable ensuring that an EV is only charged when plugged in, Equations (A19) and (A20). Furthermore, our model does not account for Vehicle-to-Grid services. Hence, we assume that an electric vehicle's battery is solely discharged during driving (A21).

This assumption implies that electric vehicles in our model offer flexibility primarily by deferring the charging process.

$$-l_{max}^{BESS} \leq l_{t,n}^{BESS} \leq l_{max}^{BESS}, \quad \forall n \in N_{BESS} \tag{A15}$$

$$l_{t,n}^{BESS} \leq l_{t,n}^{PV,used}, \quad \forall n \in N_{BESS} \tag{A16}$$

$$0 \leq SOC_{t,n}^{BESS} \leq SOC_{max}^{BESS}, \quad \forall n \in N_{BESS} \tag{A17}$$

$$SOC_{t+1,n}^{BESS} = SOC_{t,n}^{BESS} * \left(1 - \eta^{BESS}\right) + l_{t,n}^{BESS} * \Delta T, \quad \forall n \in N_{BESS} \tag{A18}$$

$$0 \leq l_{t,n}^{EV,ch} \leq l_{max}^{EV,ch} * u_{t,n}^{EV,home}, \quad \forall n \in N_{EV} \tag{A19}$$

$$0 \leq SOC_{t,n}^{EV} \leq SOC_{max}^{EV}, \quad \forall n \in N_{EV} \tag{A20}$$

$$SOC_{t+1,n}^{EV} = SOC_{t,n}^{EV} + l_{t,n}^{EV,ch} * \Delta T - l_{t,n}^{EV,drive} * \Delta T, \quad \forall n \in N_{EV} \tag{A21}$$

with:

$l_{max}^{BESS|EV,ch}$ Maximum charging and discharging rate of the BESS/EV

$SOC_{max}^{BESS|EV}$ Maximum BESS/EV capacity

$u_{t,n}^{EV_{home}}$ Binary variable that indicates that the EV is pluged in at time t

$SOC_{t,n}^{BESS|EV}$ Energy capacity of the BESS/EV at t (kWh)

$\eta^{BESS}$ Efficiency the BESS (implemented as standing losses)

In our model, we represent heat pumps as thermal storage devices with a certain degree of inertia, which is defined by constraints (A22)–(A24). A synthetic heat pump profile depicts the required electrical power to fulfill the household's heat demand. We posit that a deviation from this profile beyond a defined dead band triggers a temperature change, consequently causing charging or discharging of the thermal storage (A24). Equation (A23) outlines the SOC limits of the thermal storage, thereby determining the heat pump's load-shifting potential. Equation (A22) guarantees that the heat pump's electrical power limits are not breached.

$$-l_{t,n}^{HP,SQ} \leq l_{t,n}^{HP} \leq l_{max,n}^{HP} - l_{t,n}^{HP,SQ}, \quad \forall n \in N_{HP} \tag{A22}$$

$$0 \leq SOC_{t,n}^{HP} \leq SOC_{max,n}^{HP}, \quad \forall n \in N_{HP} \tag{A23}$$

$$SOC_{t+1,n}^{HP} = SOC_{t,n}^{HP} + l_{t,n}^{HP} * \Delta T, \quad \forall n \in N_{HP} \tag{A24}$$

with:

$l_{t,n}^{HP}$ Power that adjusts the set point of the heat pump

$l_{t,n}^{HP,SQ}$ Power demand of the heat pump to ensure the set room temperature

$SOC_{t,n}^{HP}$ Capacity of the thermal storage

*Appendix C.2. Central Optimization—Transformer Optimization Problem*

To simulate a direct-load-control tariff l, we employ a slightly different optimization problem. The objective of DLC is to alleviate grid congestion by directly manipulating the operation points of flexible assets. This is achieved by minimizing transformer overloads. Similar to the decentralized optimization model, we include a penalty term in the objective function to minimize household discomfort through load restrictions (A25). In addition, we add a term that includes the electricity costs and the revenues from PV injection to the grid to include self-consumption on the grid level. Additionally, we incorporate a term that accounts for electricity costs and revenues from PV injection to the grid to include self-

consumption at the grid level. The operation of flexible assets and the penalty terms mirror the definition of the decentralized model and are represented by Equations (A14)–(A24).

$$
\begin{aligned}
\min \sum_t^T &\mathrm{Max}\left(l_t^{\mathrm{trafo}} - l_{\max}^{\mathrm{trafo}}\right) * p^{\mathrm{VOLL}} \\
&+ \sum_t^T \sum_n^N l_{n,t} * p_t^{\mathrm{buy}} - i_{n,t} * p_t^{\mathrm{sell}} \\
&+ \sum_t^T \sum_n^N C_{t,n}^{\mathrm{EV,HO_{Pen}}}, \quad \forall t, \ n
\end{aligned}
\tag{A25}
$$

$$
l_t^{\mathrm{trafo}} = \sum_n^N l_{n,t} - i_{n,t}
\tag{A26}
$$

**Appendix D. Grid Cost Recovery**

In a post-optimization calculation, we adjust the tariff levels that were calculated during the initial calibration to achieve cost recovery. The device scheduling optimization alters the system's load profile, leading to adjustments in variable grid and energy costs. These revised costs are computed utilizing post-optimization load statistics, as outlined in Equations (A2) and (A3). Subsequently, the tariffs are recalibrated with the updated variable network costs and energy costs, following the same formulas as during the initial calibration.

**Appendix E. Impact Measures**

We assess the impact of various tariff designs across the following four dimensions:

1. **Effectiveness:** We determine how much each tariff reduces the peak load at each node, as well as the peak load of the overall system.
2. **Efficiency:** We calculate the extent to which each tariff changes the total system cost, encompassing energy costs, grid costs, and residual costs.
3. **Impact on Profitability of New Technologies:** We compute the cost difference between households equipped with and without heat pumps, electric vehicles, PV systems, or batteries across various tariff scenarios. This assessment focuses exclusively on the electricity cost implications resulting from a tariff scheme, excluding investment cost considerations. To quantify the variations in electricity costs among households with distinct appliance equipment, we employ a multiple linear regression model featuring interaction terms. Initially, we ascertain the impacts of appliance equipment on household electricity bills under a single tariff. This is achieved through Equation (A27). To evaluate changes in profitability, we extend this calculation with a dummy variable for the new tariff. The parameter estimates of this dummy variable reflect the electricity cost disparities arising from the new tariff. In simpler terms, this relationship can also be represented by the difference in the average electricity cost of all households with a particular equipment level e under a new tariff $\overline{C}_{\mathrm{hh,e}}^{\mathrm{tariff}}$ and the average electricity cost of households with the same equipment level under the existing tariff $\overline{C}_{\mathrm{hh,e}}^{\mathrm{SQ}}$ (Equation (A28)).

$$
\begin{aligned}
C_{\mathrm{hh}} = \ &\alpha + \beta_1 \mathrm{EV} + \beta_2 \mathrm{HP} + \beta_3 \mathrm{PV} + \beta_4 \mathrm{PV} * \mathrm{BESS} + \beta_5 \mathrm{PV} * \mathrm{EV} + \beta_6 \mathrm{PV} * \mathrm{HP} + \beta_7 \mathrm{PV} * \mathrm{EV} * \mathrm{HP} + \beta_8 \mathrm{HP} * \\
&\mathrm{EV} + \beta_9 \mathrm{PV} * \mathrm{HP} + \beta_{10} \mathrm{PV} * \mathrm{HP} + \beta_{11} \mathrm{PV} * \mathrm{BESS} * \mathrm{EV} * \mathrm{HP} + \epsilon_{\mathrm{hh}}
\end{aligned}
\tag{A27}
$$

$$
\Delta C_{\mathrm{hh,e}} = \overline{C}_{\mathrm{hh,e}}^{\mathrm{tariff}} - \overline{C}_{\mathrm{hh,e}}^{\mathrm{SQ}}, \quad \forall e \ \mathrm{in} \ E
\tag{A28}
$$

4. **Equity:** We assess how much each tariff influences consumer bills across different income brackets. This measure is obtained by multiplying the bill impact for each household type by the proportion of consumers from each income bracket belonging to that household type. We base this allocation on survey findings from the Swiss Household Energy Demand Survey (SHEDS) described in [27] (see Table A2). Due

to the substantial variability in electricity costs among households with varying appliance equipment, we standardize electricity costs to one consumption unit (kWh).

By evaluating these dimensions, we can gain a comprehensive understanding of the effects of diverse tariff strategies on various facets of the energy system, economic outcomes, and fairness considerations.

**Table A2.** Share of SHEDS respondents from different income brackets per household type.

| | Household Features | | | | Income Class CHF per Month | | | | | |
|---|---|---|---|---|---|---|---|---|---|---|
| HH-Type | HP | EV | BESS | PV | 3000–4499 | 3000–4499 | 4500–5999 | 6000–8999 | 9000–12,000 | >12,000 |
| type1 | 0 | 0 | 0 | 0 | 4% | 9% | 19% | 31% | 20% | 16% |
| type2 | 1 | 0 | 0 | 0 | 3% | 5% | 13% | 27% | 29% | 24% |
| type3 | 0 | 1 | 0 | 0 | 2% | 4% | 8% | 22% | 32% | 32% |
| type4 | 0 | 0 | 0 | 1 | 10% | 5% | 15% | 32% | 21% | 17% |
| type5 | 1 | 1 | 0 | 0 | 4% | 0% | 8% | 23% | 27% | 38% |
| type6 | 1 | 0 | 0 | 1 | 2% | 10% | 11% | 22% | 21% | 33% |
| type7 | 0 | 1 | 0 | 1 | 0% | 0% | 10% | 10% | 40% | 40% |
| type8 | 0 | 0 | 1 | 1 | 10% | 5% | 15% | 32% | 21% | 17% |
| type9 | 1 | 1 | 0 | 1 | 0% | 5% | 5% | 16% | 21% | 53% |
| type10 | 1 | 0 | 1 | 1 | 2% | 10% | 11% | 22% | 21% | 33% |
| type11 | 0 | 1 | 1 | 1 | 0% | 0% | 10% | 10% | 40% | 40% |
| type12 | 1 | 1 | 1 | 1 | 0% | 5% | 5% | 16% | 21% | 53% |

**Appendix F. Load Profiles**

For the diverse appliances, synthetic load profiles were generated. To capture the diversity of households' behaviors, an ensemble of 100 synthetic load profiles was initially created, from which 300 load profiles were selected for the simulation. In the subsequent part of this section, the process of generating synthetic profiles for each simulated appliance is detailed.

Regarding the non-flexible household load, an ensemble of 100 synthetic load profiles was generated using the LoadProfileGenerator (LPG) [18]. These profiles were scaled to match the average measured electricity consumption of households within the grid area of a distribution grid operator in the canton of Aargau in 2018, amounting to 4873 kWh per household. This procedure results in 100 diverse household load profiles, each representing the non-flexible electricity consumption of households, in a one-hour resolution.

The synthetic load profiles for electric vehicle charging were also generated using the LPG. A matching electric vehicle charging profile was assigned to each household load profile. We assumed a charging power of 11 kW, an average vehicle battery capacity of 50 kWh, and an average consumption of 16 kWh per 100 km. We also assumed that load shifting for EV charging would not compromise households' comfort if the EV was fully charged at the desired departure time.

The load profile for HP was generated by determining the relative heating demand for each hour of the year and scaling it by the thermal energy demand of households. The latter was estimated using the measured gas consumption of households within the grid area of a distribution grid operator in the canton of Aargau in 2018. The resulting mean load of the heat pump profiles is $\bar{l}_{t,hp}^{max} = 2.75$ kW. To simulate the flexibility operation of the heat pump, it was assumed that a load shift of 2 h would only marginally impact the perceived room temperature, and therefore, would not affect the household's comfort.

For PV profiles, more than 300 standard generation profiles with varying module orientation and slope were calculated. The distribution of module orientation and slope

was assigned based on realized PV plants in the considered distribution grid. This approach introduces heterogeneity in the generation patterns to the simulation model. The synthetic generation profiles were then allocated to households and sized according to the realized PV plants in a specific distribution grid area in Switzerland. The peak power of the PV plants was adjusted to cover the household load.

Only households that were already equipped with PV could be additionally equipped with a BESS. To simulate the operation of a BESS, the following assumptions were made: the battery capacity is sized with a factor of 1.5 relative to the PV peak power, the maximum charging power was assumed to be 3 kW, and the roundtrip efficiency was set at 0.9.

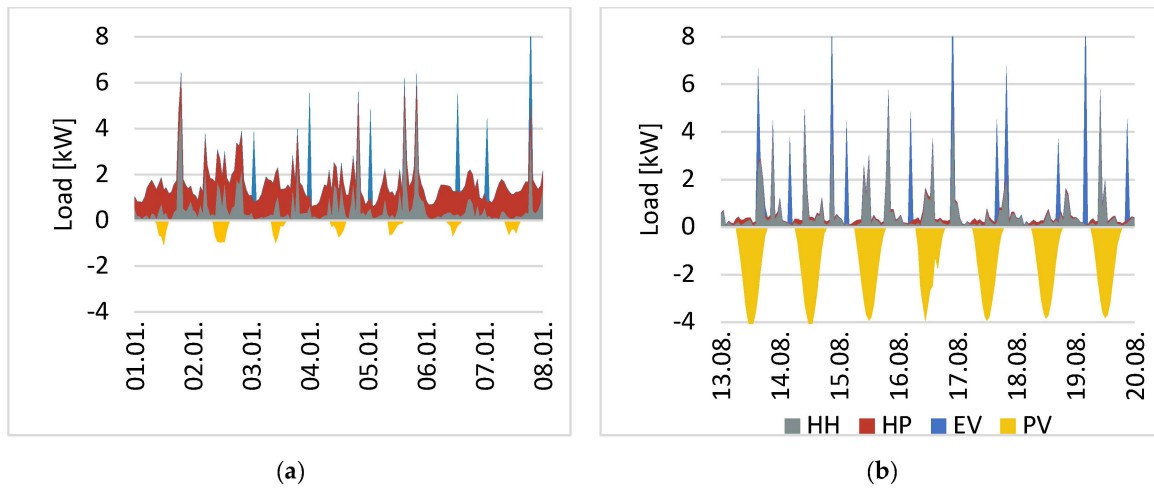

(**a**)  (**b**)

**Figure A1.** Synthetic load profiles for one household over one week in (**a**) winter and (**b**) summer.

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
