# Peer review of "Design and Impact of Grid Tariffs"

_energies, doi:10.3390/en17061364_

Round 1

Reviewer 1 Report

Comments and Suggestions for Authors

The paper aims to analyze the effect of capacity tariff and novel continuous tariff signal on the grid load with a comparison between their performance to the performance of another approaches in term of avoiding rebound peak. The topic in interesting specially with the rise of automated and electrification of many economic sector. The paper is well structured. My comments are.

1-       All figures are not clear! Please improve the quality and add a small explanation under each figure or in the text.

2-       Check line 142: reference is not well inserted.

3-       Line 221, please change the number of the figure! It should be figure 5

4-       Line 321: correct the number of the figure from 1 to 10

5-       In Figure 2: please use the same notation! In the text, authors mentioned capacity charge and, in the figure, power

6-       Regarding figure 3: authors are requested to explain the different structural approaches in terms of advantages and disadvantages.

7-       In the line 122: authors mentioned that they will study the effect of 9 tariff scenarios, however, they presented only 8.

8-       In the methodology part, authors are required to present the mathematical equations used.

9-       In lines 160-161: regarding the estimation of appliance equipment for the year 2050, the prediction is based on households lacking flexible  technologies adopt at least one flexible technology. Could the authors add more details about it!

10- Table 1: authors are required to describe the results obtained in the table

11- In paragraph 5.4.1: is there any effect of size of household?

12- In the discussion part: authors are required to compare the results between 2020 and 2050.

Comments on the Quality of English Language

minor editing

Reviewer 2 Report

Comments and Suggestions for Authors

please the review

Comments on the Quality of English Language
